# Rosetta design with co-evolutionary information retains protein function

**Samuel Schmitz**[1☯], **Moritz Ertelt**[2,3☯], **Rainer Merkl**[3], **Jens Meiler**[1,2]*

**1** Department of Chemistry, Vanderbilt University, Nashville, Tennessee, United States of America,
**2** Institute for Drug Discovery, Leipzig University, Leipzig, Germany, **3** Institute of Biophysics and Physical Biochemistry, University of Regensburg, Regensburg, Germany

☯ These authors contributed equally to this work.
* jens@meilerlab.org

**Data Availability Statement:** All relevant data are within the paper and its Supporting Information files.

## Abstract

Computational protein design has the ambitious goal of crafting novel proteins that address challenges in biology and medicine. To overcome these challenges, the computational protein modeling suite Rosetta has been tailored to address various protein design tasks. Recently, statistical methods have been developed that identify correlated mutations between residues in a multiple sequence alignment of homologous proteins. These subtle inter-dependencies in the occupancy of residue positions throughout evolution are crucial for protein function, but we found that three current Rosetta design approaches fail to recover these co-evolutionary couplings. Thus, we developed the Rosetta method ResCue (residue-coupling enhanced) that leverages co-evolutionary information to favor sequences which recapitulate correlated mutations, as observed in nature. To assess the protocols via recapitulation designs, we compiled a benchmark of ten proteins each represented by two, structurally diverse states. We could demonstrate that ResCue designed sequences with an average sequence recovery rate of 70%, whereas three other protocols reached not more than 50%, on average. Our approach had higher recovery rates also for functionally important residues, which were studied in detail. This improvement has only a minor negative effect on the fitness of the designed sequences as assessed by Rosetta energy. In conclusion, our findings support the idea that informing protocols with co-evolutionary signals helps to design stable and native-like proteins that are compatible with the different conformational states required for a complex function.

## Author summary

In homologous proteins, functionally or structurally important residues are strongly conserved. Thus, the consideration of conservation signals during protein design protocols can help to create sequences that are more native-like. However, the number of conserved residues is small in many proteins and not all important residues can be captured by conservation analysis. Residues are forming networks whose composition is dictated by protein structure function and thus is visible through the co-evolutionary analysis. Nowadays, advanced methods allow us to deduce these networks from multiple sequence alignments. Thus, we have implemented the novel Rosetta method termed 'ResCue' that

**Funding:** This work was supported by National Institute of Allergy and Infectious Diseases (NIAID) U19-AI117905-01 and by National Institute of Health S10OD023680. The funders had no role in study design, data collection and analysis, decision to publish, or preparation of the manuscript.

**Competing interests:** The authors have declared that no competing interests exist.

informs the design protocol with co-evolutionary signals. Recapitulation designs based on ten difficult benchmarks made clear that this protocol creates sequences that are more native-like than three other, state-of-the-art design protocols.

This is a *PLOS Computational Biology* Methods paper.

## Introduction

Proteins play a vital role in fundamental processes of life, and their diverse three-dimensional structures allow for highly diverse functions. Computational protein design explores the sequence landscape and side chain conformational space for a given protein backbone to find a residue combination that supports a function. The protein modeling suite Rosetta [1] has been applied with marked success on various applications [2, 3], including protein [4] and enzyme design [5]. A critical element of Rosetta is a scoring function that is fine-tuned to respect knowledge-based statistics and physical approximations. Without additional restraints, this scoring function reflects the thermodynamic stability of one static protein conformation in a distinct environment [6].

However, protein function often relies on structural flexibility [7], thus multiple Rosetta protocols have been developed to favor sequences which do not only thermostabilize but also account for protein flexibility. Multi-state design (MSD), for example, supports design on multiple protein conformations simultaneously which benefits the design of conformational changes [8–10]. The MSD implementation RECON [8, 11] optimizes in an iterative protocol the individual sequences of the conformational states. Each iteration increases a restraint to converge the individually designed sequences into a single sequence that supports all conformations.

Improving thermodynamic stability or function of a given protein is an important aspect of protein design [12]. As protein sequences observed in nature are often close to the optimum [13], the design of sequences constrained towards native conformations and sequences is a successful strategy. It can be implemented by using sequence profiles [14] that mirror the residue occupancy at each position of a backbone and serve as additional constraints on sequence selection. However, as each residue is treated independently, a severe limitation of sequence profile design is the neglection of subtle interdependencies between residue occupancies.

The reasons for these mutual dependencies are often the maintenance of structural stability by compensatory mutations but are also more importantly related to sophisticated functional aspects like information transmission, conformational plasticity, and the binding of ligands or other proteins [15, 16]. Thus, a network of evolutionary constraints may exist in a protein that fine-tunes the occupancy of several pairs of residue-positions. Various methods like GREM-LIN [17], plmDCA [18], and PSICOV [19] have been developed to identify these constraints, which are also named couplings, to indicate the dependency between the occupancy of residue pairs. In a pioneering study, co-evolutionary fitness landscapes have been used to design three different stable protein folds with the ability to bind native ligands with high affinity [20].

Pairwise sequence requirements in natural proteins are a consequence of maintaining thermodynamic stability, structural flexibility (plasticity), and other requirements for protein function, such as recognizing interaction partners, catalyzing chemical reactions, and many more. Computational protein design with Rosetta primarily favors thermodynamic stability and is conceptually unaware of couplings required for protein flexibility and/or function. The premise of this study is that this restriction in evolutionary tolerated sequence space is not reflected

in Rosetta designed proteins. This leads to design solutions that are thermodynamically stable but might change flexibility or lose function. While custom protocols for a specific design task can circumvent this shortcoming, we wondered about a general approach to maintain native-like couplings in the sequences designed beyond the couplings dictated by thermodynamic stability. For this study, we evaluate a number of computational design protocols in Rosetta: 1) One biased towards the wild-type sequence as a baseline for comparison, 2) Design with a sequence profile, which encodes the sequence space as observed in functional proteins, 3) RECON multi state design, which has the potential to capture couplings critical for protein plasticity, and 4) Constraining co-evolving residues directly in the Rosetta design process.

We hypothesize that incorporating evolutionary constraints in the Rosetta design process will allow us to optimize the sequence across all functionally relevant conformations even for single state design (SSD), including intermediate states that are difficult to obtain experimentally [21]. Thus, we have implemented a novel RosettaScripts [22] element, the ResCue (residue coupling enhanced) mover, which transforms coupling strengths inferred from a MSA into an energy function bias (restraint). These restraints are generalizable and applicable on different design scenarios that can be addressed with Rosetta. To evaluate our method, we captured two performance metrics: First, we measured the recovery of couplings. Second, we assessed the overall sequence recovery of the full protein and of residues which were reported as functionally relevant in literature. We found that proteins designed with ResCue had significantly higher recovery rates compared with three other state-of-the-art design approaches.

We use native sequence recovery as one of our metrics of design success in order to facilitate comparison with other studies. Although it might appear counter-intuitive to use this measure to assess coupling recovery, we argue that it is a useful metric as increased coupling recovery will imply increased sequence recovery. Our method achieves high recovery rates by conserving networks of co-evolving residue pairs, in contrast to an alternative approach that trivially increases sequence recovery rates by limiting the number of mutations. We show, that our method is superior in recapitulating the wild-type residues especially in functionally active sites compared to other approaches and thus is suitable to retain function during design.

## Results and discussion

### Assembling a benchmark *bench*<sub>*coev*</sub> of ten proteins representing conformational flexibility

In order to test our hypothesis that co-evolutionary information helps to improve the protein design process, we assembled a benchmark of ten proteins, which we named $bench_{coev}$ (Table 1). We chose the proteins based on four criteria. First, two conformations had to be available in the Protein Data Bank (PDB), representing two functionally different states e.g. without or with a bound substrate. Second, we accepted only structures with an experimental resolution of 3 Å or better. Third, for each protein of length *N*, we confirmed that 10 x *N* homologous sequences were available in databases, which is a prerequisite for a reliable determination of coupling [23]. Forth, we preferred proteins that are functionally well studied and understood. We ended up with a diverse set of two calcium binding proteins, two GTP binding proteins, one DNA binding protein, one phosphate binding protein, one enzyme, one bacterial solute binding protein and one protein that is part of an ABC transporter.

### The ResCue mover and its energy term

Our method aims at the conservation of co-evolutionary networks during the design of protein sequences (Fig 1). For their identification, we opted for GREMLIN [17] that deduces from an

**Table 1. Characterization of the ten benchmark proteins (*bench_coev*) used in this study.**

| Description | PDB IDs | Resolution [Å] | Length | RMSD [Å] | MSA size |
|---|---|---|---|---|---|
| HPPK | 1HKA<br>1Q0N | 1.5<br>1.3 | 435 | 0.5 | 4534 |
| FixJ | 1D5W 1DBW | 2.3<br>1.6 | 126 | 0.5 | 38021 |
| RasH | 6Q21<br>4Q21 | 2.0<br>2.0 | 189 | 0.5 | 46795 |
| G-protein Arf6 | 1E0S<br>2J5X | 2.3<br>2.8 | 174 | 1.0 | 36036 |
| S100A6 | 1K9P<br>1K9K | 1.9<br>1.8 | 90 | 1.9 | 14768 |
| Calmodulin | 1CKK<br>1CFD | NMR<br>NMR | 148 | 9.9 | 13561 |
| LAO Binding protein | 2LAO<br>1LAF | 1.9<br>2.1 | 238 | 4.5 | 23810 |
| Phosphate-binding protein | 1QUK<br>1OIB | 1.7<br>2.4 | 321 | 2.9 | 5898 |
| Thioredoxin reductase | 1TDE<br>1F6M | 2.1<br>3.0 | 316 | 6.6 | 33408 |
| Adenylate kinase | 1AKE<br>4AKE | 2.0<br>2.2 | 214 | 7.0 | 30589 |

## ResCue constraints

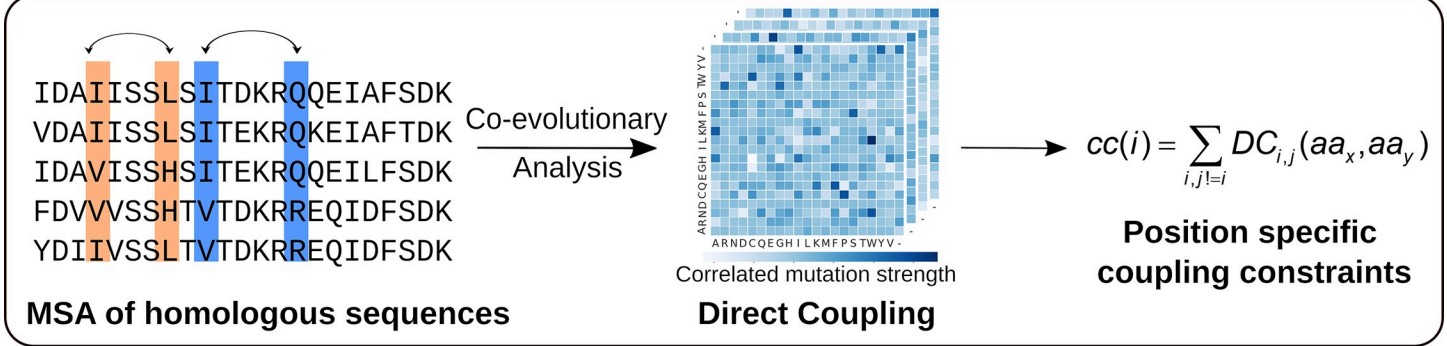

$$cc(i) = \sum_{i,j \, != \, i} DC_{i,j}(aa_x, aa_y)$$

**Position specific coupling constraints**

## Rosetta Sequence Design

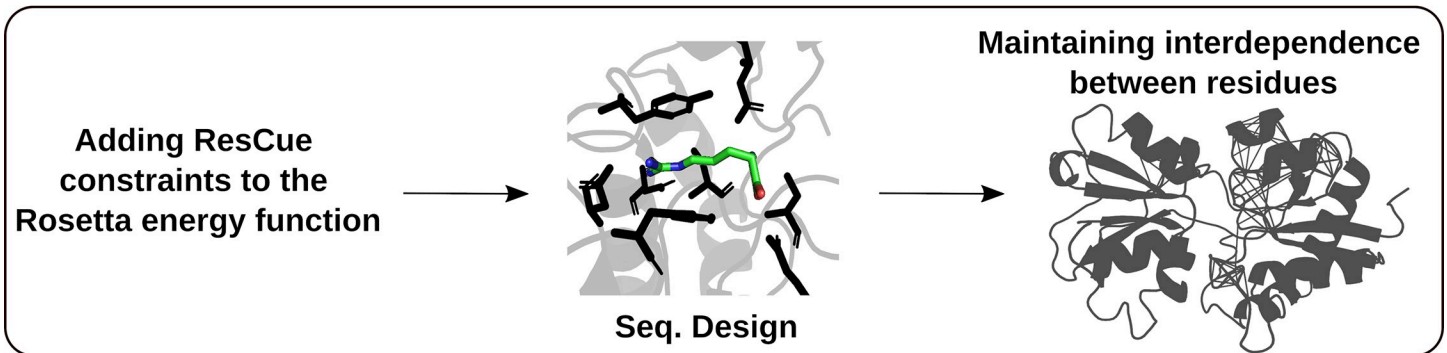

**Fig 1. Basic concept and application of ResCue.** Co-variance scores are deduced from an MSA of homologous sequences and converted to position specific coupling constraints $cc(i)$. The ResCue constraints are then added to the Rosetta energy function to maintain the interdependencies between residues while protein sequence design.

MSA of homologous sequences a four-dimensional coupling tensor storing the co-evolutionary scores. The first two dimensions list two residue positions and the last two indicate their amino acid interdependencies for all possible combinations. Large positive values represent a strong coupling and negative values indicate their incompatibility; but most values are close to zero. The tensor allows us to quickly deduce a score (Eq 1) for the strength of the coupling constraint $cc(i)$ for each individual residue $i$. These scores are then used to constrain sequence design as an add-in to the Rosetta energy function. Note that it is essential to balance carefully between having an efficient constraint but not over-writing the standard energy function, since the designed proteins should fulfill the coupling restraints *and* be physically realistic.

## Sophisticated design protocols sample sequences of higher energy

To evaluate our method, we assessed the performance of four established design protocols on the protein benchmark set *bench_coev*: a) An unmodified, default Rosetta single-state design protocol that served as a reference (RoSSD) [1], b) RECON MSD design, c) SeqProf Mover (SeqProf), which is sequences profile design using a position-specific scoring matrix (PSSM) [24], d) biased design to prefer the native sequence (FavorNative), and e) our co-evolutionary informed design ResCue. The same MSA was used to derive the PSSMs for SeqProf and the residue-specific coupling constraints $cc(i)$ (Eq 1) utilized with ResCue.

As noted above, it is essential, to balance carefully between having an efficient coupling restraint and designing physically realistic sequences as dictated by the Rosetta energy function. By restraining Rosetta to bias the sampling towards a desired goal, the energy landscape is modified. As a result, when reevaluating the solutions with the unmodified energy function, the energy can and often does increase (get worse). The ResCue protocol (S2 Supplement) was parametrized to produce designs with Rosetta energies comparable to the established SeqProf method and substantially increased coupling recovery. In order to assess the energy increase of the different design approaches, we determined the difference of the Rosetta total energy to the relaxed wild-type structure with the best energy, normalized by protein length. As expected, all design approaches with constraints had significantly higher Rosetta energies compared to the relaxed wild type (Fig 2), (Mann–Whitney U test (MW) p < 1.0e-04 for all three comparisons). The differences per residue were -0.15 ± 0.11 REU for single state design, +0.55 ± 0.76 REU for RECON MSD, + 0.28 ± 0.14 REU for the SeqProf design, -0.057 ± 0.15 REU for FavorNative, and +0.13 ± 0.15 REU for our ResCue mover. As the latter value is significantly lower than that of the RECON protocol (MW p = 6.4e-195) and comparable to the SeqProf design, we concluded that our concept of considering evolutionary constraints affects the scoring function less than a well-established MSD approach. The FavorNative design energies remain on average close to the wild-type energies (-0.057 REU). Most likely, for several positions residue choices with similar energies exist and FavorNative helps to select the native residue. As expected, sequence recovery increases substantially with the FavorNative method, which is a trivial result as the correct solution is input into the method. Thus, this approach would not allow the design of new sequences that retain structure, plasticity, and function. Details on the execution of each experiment and the used constraint weights can be found in S2 Supplement.

## ResCue recovers networks of co-evolving residues

Having shown that our scoring of couplings had no drastic effect on sequence energies, we assessed the conservation of couplings by analyzing for each benchmark *prot* the designed sequence $seq_{Design}$ and the native sequences $seq_{Native.}$ We determined the coupling recovery score $crs(prot)$ (Eq 3), which quantifies how well the inferred residue interaction network was maintained. To compute this score, we first calculated for each $seq_{Design}$ the sum of the

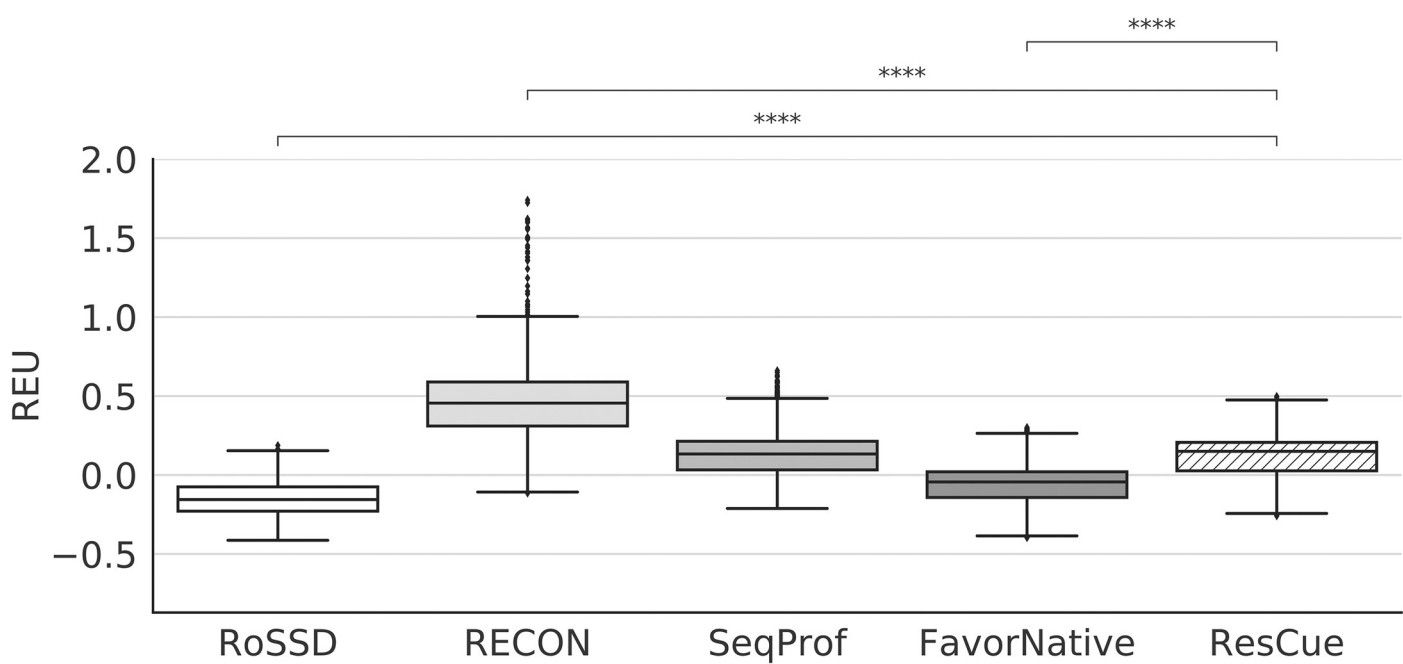

**Fig 2. Distribution of Rosetta total energies for the full benchmark design.** Energies are given in Rosetta energy units (REU) relative to the wild-type on a per residue basis for the five design protocols. For this figure and all subsequent boxplots, the median is indicated as a black line; boxes depict the interquartile range (IQR), whiskers represent 1.5 x IQR. Results of a two-sided Mann-Whitney-Wilcoxon test are indicated as follows: * ≙ 1.00e-02 < p < = 5.00e-02, * ≙ 1.00e-03 < p < = 1.00e-02, * ≙ 1.00e-04 < p < = 1.00e-03, *** ≙ p < = 1.00e-04.

corresponding scores $cc(i)$. One can consider the cumulative strength of pairwise couplings as a measure for the selective functional pressure on a particular residue $i$ [16]. For normalization, the resulting $cs(seq_{Design})$ value (Eq 2) was divided by $cs(seq_{Native})$. Note that $crs(prot)$ can assume values greater than one.

Fig 3A depicts the $crs$ values of the sequences designed with the five protocols; the standard deviation was ≈ 10% in all cases. The unconstrained Rosetta protocol RoSSD reached an average $crs$ value of 21%. The performance of RECON MSD was slightly better with an average $crs$ value of 25% and SeqProf design gained an average $crs$ value of 28%. FavorNative reached an average $crs$ value of 52% with a higher standard deviation compared to other protocols with 18.3%. In contrast, ResCue reached an average $crs$ value of 109%. Note, that the $crs$ value can be larger than 100% which would suggest that the designed sequences fulfill additionally restraints not found in the native sequences $seq_{Native}$, but in homologs.

These data show that the native Rosetta protocol is not suitable to completely recover the evolutionary constraints for functional connectivity between residues across our ten proteins. Moreover, we expected a better performance of RECON, since multistate design optimizes over both conformations at the same time. This optimization should consider residue couplings that are in spatial proximity in either state. As expected, SeqProf failed to drastically improve the performance as mutual residue-dependencies are ignored. In contrast, the average $crs$ value of ResCue exceeds 100%.

### Preserving evolutionary constraints by means of ResCue improves native sequence recovery and sequence similarity

As further quality measures, we determined native sequence recovery $nsr(seq_{Design})$ (Eq 4) and sequence similarity $seqsim(seq_{Design})$ (Eq 6) values by comparing the designed sequences and

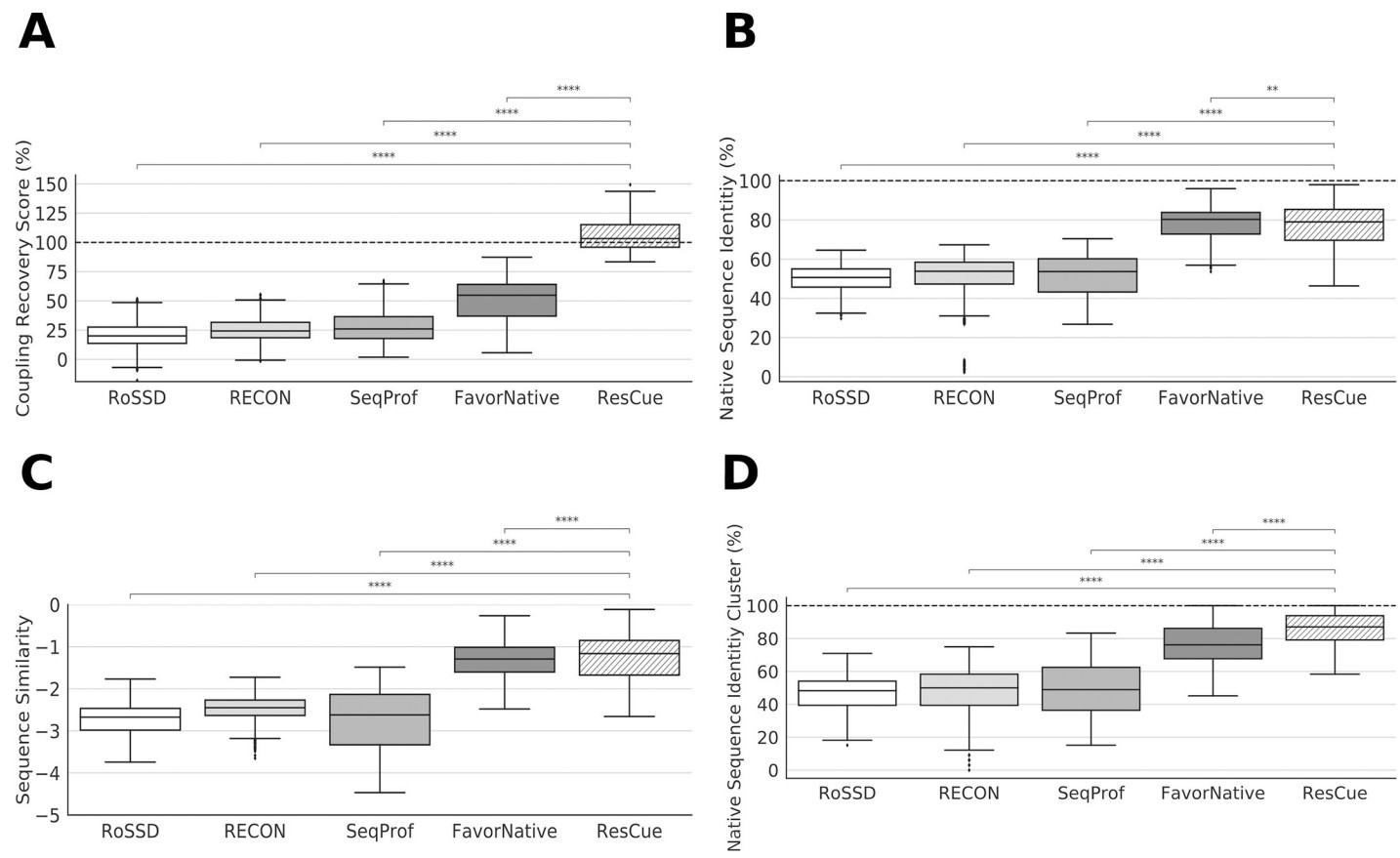

**Fig 3. Performance of four different design approaches.** (**A**) Coupling recovery scores *crs* deduced for all designed sequences. The 100% value is indicated by the dashed line. (**B**) Native sequence recovery *nsr* of the four design approaches. (**C**) Sequence similarity *seqsim* of the full-length sequences. (**D**) Native sequence recovery *nsr$_{CN}$* of residues found in clustered networks. The scores are summarized with boxplots as explained above. Results of a two-sided Mann-Whitney-Wilcoxon test are indicated as follows: * ≙ 1.00e-02 < p < = 5.00e-02, * ≙ 1.00e-03 < p < = 1.00e-02, * ≙ 1.00e-04 < p < = 1.00e-03, *** ≙ p < = 1.00e-04).

the native ones (Fig 3B and 3C). RoSSD reached an average *nsr* value of 50 ± 6.3% and a *seqsim* value of -2.72 ± 0.34. For RECON, the *nsr* and *seqsim* values were 50 ± 12.2% and -2.50 ± 0.35, respectively. SeqProf design gained an averaged *nsr* value of 51 ± 10% and a *seqsim* value of -2.74 ± 0.35. FavorNative reached *nsr* and *seqsim* values of 78 ± 9.15% and -1.28 ± 0.5. Note that the FavorNative weights were tuned to approximate the ResCue sequence recovery. Our ResCue design showed a significant increase both in the *nsr* and the *seqsim* values, which were 78 ± 11.7% and -1.20 ± 0.61, respectively. Compared to the other design approaches, these improvements were statistically significant (MW *p* < 5.0e-04 for *nsr* and *seqsim*).

We wanted to know, whether these protocol-specific performance differences in *nsr* and *crs* values affect each individual protein of the benchmark *bench$_{coev}$* and determined the *nsr* improvements and the *crs* improvements. For each of the two conformations of a protein, the *nsr* value of RoSSD was subtracted from the *nsr* value reached by each of the three other design protocols, namely RECON, SeqProf, and ResCue. Analogously, the *crs* values were processed. Thus, a difference greater than zero indicates an improvement compared to RoSSD, whereas a value smaller than zero indicates that the protocol performed worse than the reference.

In Fig 4, these pairs of values are plotted for each protein of *bench$_{coev}$*. RECON showed a slight increase except for the two conformational states of the thioredoxin reductase (two data-points in the lower left quadrant). Most likely, these results are due to the low resolution of one

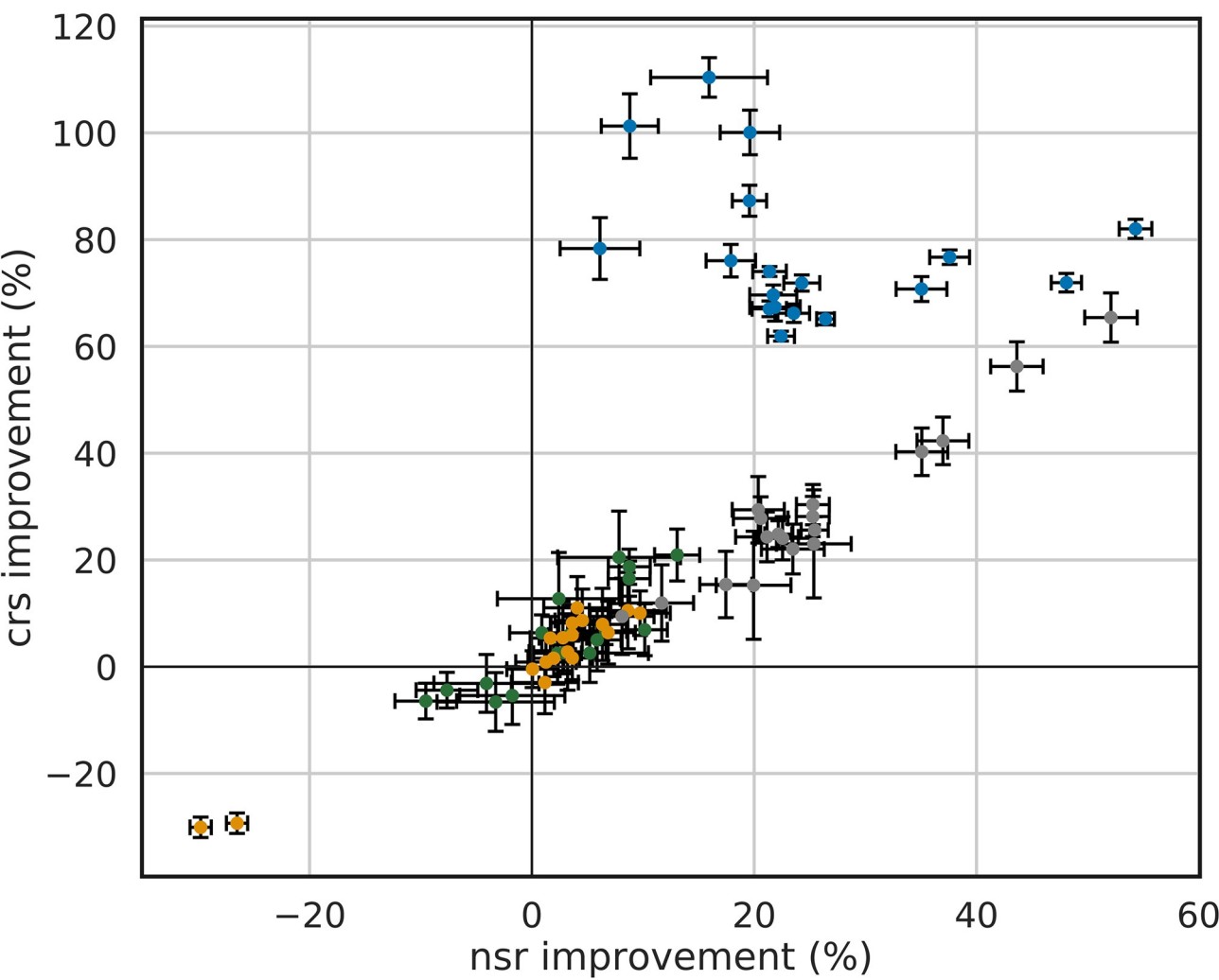

**Fig 4. Improvement of native sequence recovery values and coupling recovery scores for the two conformational states of all ten benchmark proteins.** For each state, the *nsr* value of the default Rosetta protocol was subtracted from the *nsr* value reached by one of the other protocols. The *crs* values were processed analogously. The following color code indicates the design protocols: RECON (orange), SeqProf (green), ResCue (blue), and FavorNative (gray).

state, which was 3.0 Å. SeqProf slightly improved sequence identity and coupling recovery values in six of the proteins (thioredoxin reductase, LAO binding protein, phosphate-binding protein, S100A6, FixJ and HPPK) and impaired them in four proteins (RasH, G-protein Arf6, calmodulin and the adenylate kinase). In contrast, ResCue design improved *nsr* and *crs* values for the full benchmark set.

Taken together, ResCue directed the design process towards native sequences that facilitate the coupling of residues. Considering these constraints comes at the expense of a relatively moderate energy increase, which however, is not energetically more expensive than maintaining sequence composition by means of sequence profiles.

## ResCue recovers functionally relevant residues

Residues involved in evolutionary couplings often form networks [7, 25, 26] and the *cc(i)* values are a measure for the selective functional pressure on a particular residue [16]. To study

the most prominent cases, we adopted a previous approach [25] and identified all residues ($res_{cc}^{20}(prot)$) having a coupling constraint $cc(i)$ above the 20$^{\text{th}}$ percentile. As expected, these residues were often described as functionally relevant in the literature (see discussion for individual proteins below). We used the residues to compute coupling networks (CN) and determined for the corresponding sets of residues the $nsr_{CN}$ values, which were higher than the global sequence recovery $nsr(prot)$ values. The average $nsr_{CN}$ value of the RoSSD protocol was 46 ± 8.8%, for RECON, SeqProf, and FavorNative design the values were 47.5 ± 14.8%, 50 ± 14.9%, and 49 ± 14.9% respectively. Our ResCue design reached an average $nsr_{CN}$ value of 87 ± 8.8% (Fig 3D), which is statistically significantly higher than that of the second-best protocol, namely SeqProf (MW $p$ = 1.9 e-22).

To study the coupling networks in detail, we present results for four benchmark proteins, LAO, FixJ, RasH, and calmodulin, which we have chosen for the following reasons: First, the binding site residues of LAO play an essential role in stabilizing both the closed and the open state. Second, FixJ residues are involved in dimerization. Third, RasH uses two highly flexible switch domains to bind GTP. Fourth, residues crucial for peptide binding in calmodulin are only moderately conserved, which complicates efforts to recognize and recapitulate them. Detailed information about the remaining benchmark proteins is provided in S1 Supplement.

## The substrate induced conformational change of the lysine-arginine-ornithine binding protein LAO

LAO is a periplasmic protein capable of binding the amino acids L-arginine and L-histidine [27]. Periplasmic transport systems consist of a substrate-binding protein and a membrane-bound complex that translocates the substrate from the periplasm to the cytoplasm. Following substrate binding, LAO undergoes a conformational change, bringing the two domains into a closed configuration that completely buries the ligand. Recently, residues crucial for substrate binding were identified [28] by performing alanine scanning and categorized into different groups: Main chain binding (D161, S72, R77), guanidino binding (D11), side chain binding (Y14, F52) and water-mediated binding (D30, S70).

Our analysis revealed that the $res_{cc}^{20}(LAO)$ residues form two networks located close to the binding site, a smaller network consisting of seven residues and a more complex one with 35 residues (Figs 5A and 6A). Four of the eight crucial residues (Y14, F52, S70, R77) are part of the more complex network, which highlights that certain combinations of binding site residues enables them to bind the ligand cooperatively. Comparing the structure with and without bound ligand showed that all $res_{cc}^{20}(LAO)$ residues adopt alternative configurations in the open and the closed configuration. Analyzing the sequence space for LAO designs showed that each of the five design approaches was able to recover the native amino acids at position 11 (aspartate) and position 30 (aspartate) (Fig 7A). Both RoSSD design and RECON failed to sample the A77. In contrast, Y14, F52, S70, and S72 were only recovered in ResCue designs. Superimposing the native structure with a ResCue design sampling the correct amino acids illustrates the similarity except for two side-chain conformations (Fig 8A).

The coupling strength $cs(seq)$ of the chosen set of residues for each protein was calculated and visualized as a bar plot. Compared to the native sequence, the coupling strength of ResCue was on average 103±50%, followed by FavorNative with 35±55%, SeqProf with 23±58%, RECON with 20±63%, and RoSSD with 4±56% (S4 Supplement). The improvement of ResCue was statistically significant compared to all other design methods (MW p < 5.0e-04) The increased sequence recovery of the ResCue protocol can therefore be attributed to the collective interaction of couplings. Full-length sequence logos of the ResCue design are provided in S3 Supplement.

## Conformational changes induced by the phosphorylation of the FixJ receiver domain

FixJ is a two-component system crucial for the symbiotic nitrogen fixation in *Sinorhizobium meliloti* [29, 30]. The FixJ receiver domain is arranged in two domains, and phosphorylation

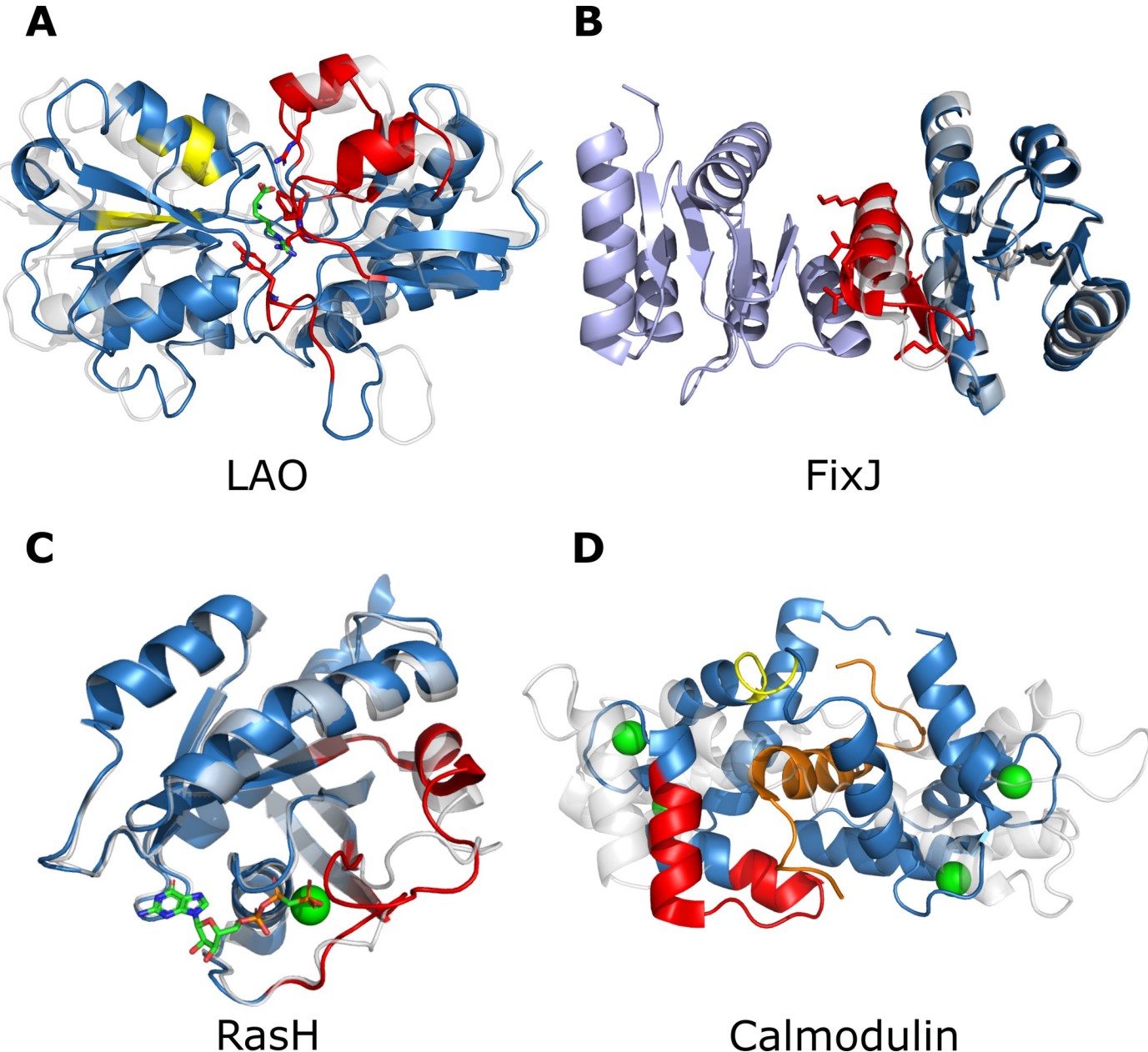

**Fig 5. Localization of highly coupled residues in four benchmark proteins.** (**A**) Network analysis of highly coupled residues mapped on the structure of LAO (PDB ID: 2LAO (unbound), 1LAF (bound)). Superposition of the unbound (grey) and the bound state (blue). The substrate L-asparagine is shown as sticks. The two networks are highlighted in red and yellow. Residues known to be crucial for substrate binding and belonging to a network are shown as sticks. (**B**) Network of highly coupled residues (red) mapped on the structure of FixJ (PDB ID: 1DBW (unphosphorylated), 1D5W (phosphorylated)). Superposition of the unphosphorylated (grey), the phosphorylated protein (blue) and a second FixJ (light blue) belonging to the dimer. Residues critical for dimerization are shown as sticks. (**C**) Network of highly coupled residues (red) mapped on the structure of RasH (PDB ID: 4Q21 (GDP bound), 6Q21 (GTP bound)). Superposition of the GDP bound state (grey) and the GTP bound state (blue). The substrate is shown as sticks. Bound magnesium is depicted as green spheres. (**D**) Network analysis of highly coupled residues mapped on the structure of calmodulin (PDB ID: 1CFD (without Ca2+), 1CKK (with Ca2+)). Superposition of the unbound (grey) and the bound state (blue). The peptide CaMKK is shown in orange. The two networks are highlighted in red and yellow.

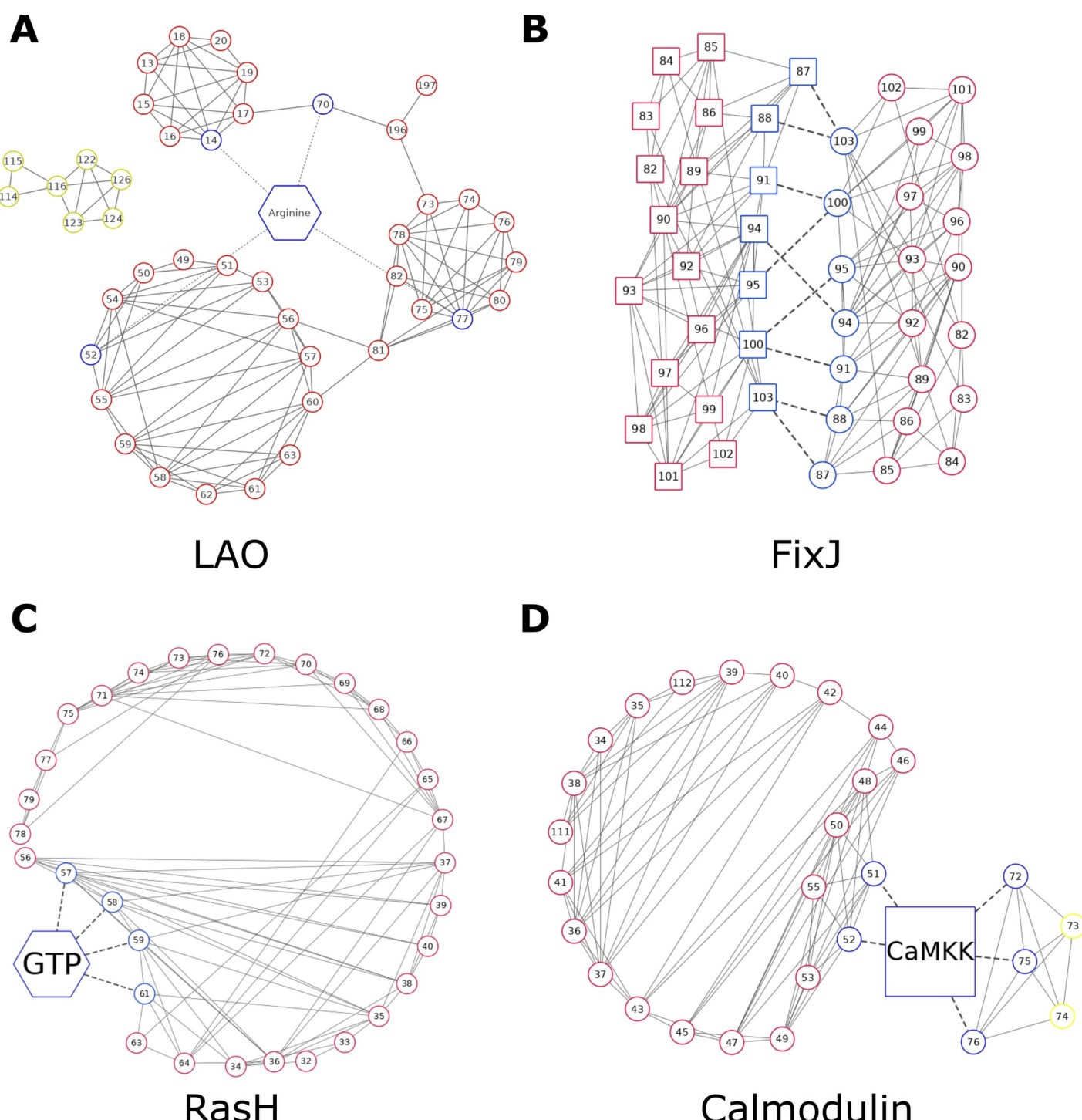

**Fig 6. Representation of residue interaction networks.** Intra-protein couplings are depicted as solid lines and dashed lines indicate substrate-binding residues or inter-protein couplings. (**A**) LAO possesses two interaction networks (red, yellow). Residues crucial for binding the substrate L-arginine are marked blue. (**B**) Residue interaction network of FixJ. Residues that are crucial for dimerization are highlighted in blue. Circles/squares distinguish coupled residues belonging to the two protomers of the dimeric complex. (**C**) Residue interaction network of RasH. Residues that are crucial for GTP hydrolysis are highlighted in blue. (**D**) Residue interaction networks (red, yellow) of calmodulin. Residues that are crucial to the binding of the peptide CaMKK are highlighted in blue.

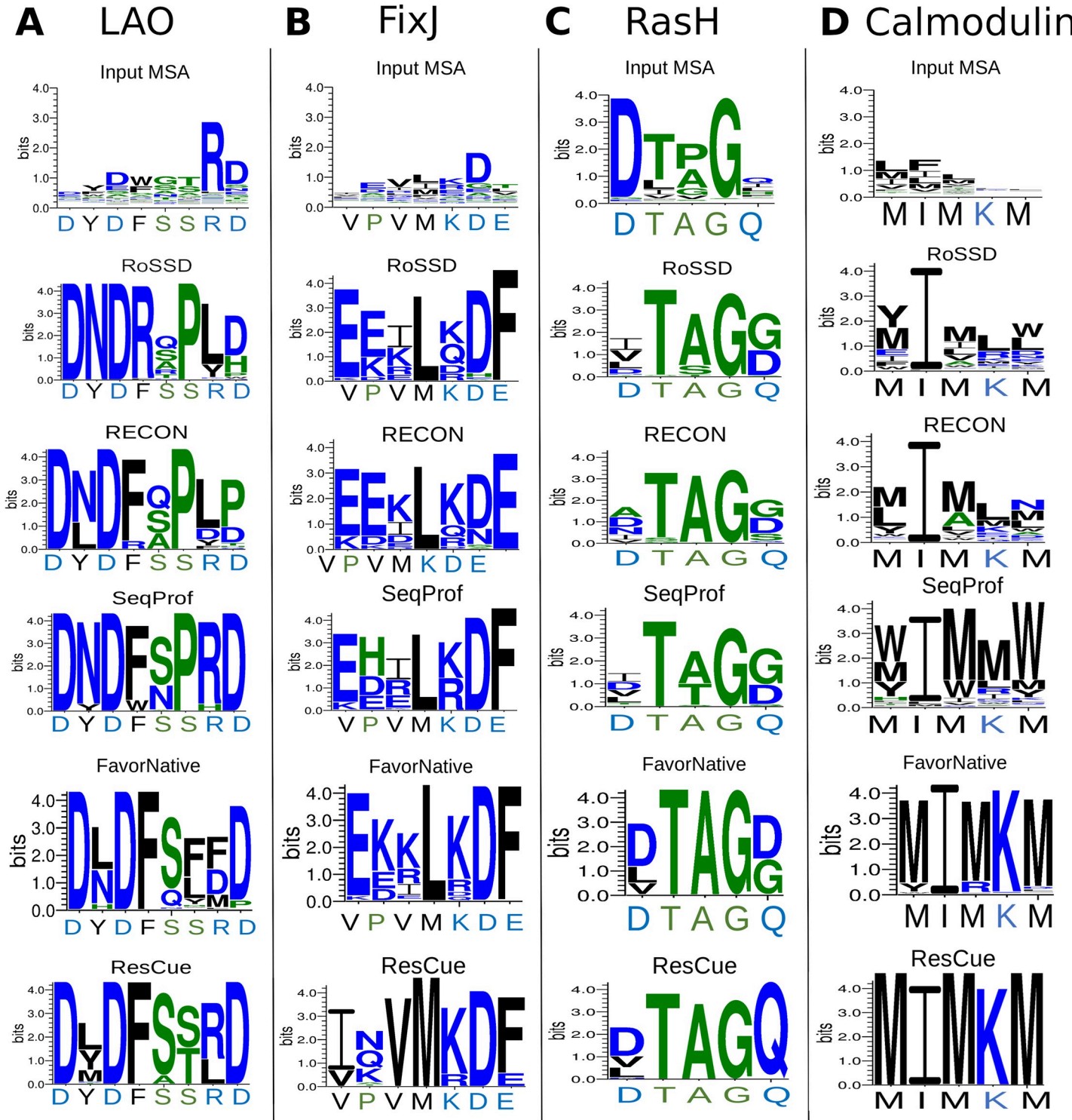

**Fig 7. Sequence logos resulting from five design protocols.** The native sequences are listed below the logos. (**A**) LAO binding site, eight residues. (**B**) FixJ dimer interface, seven residues (**C**) RasH binding site, five residues. (**D**) calmodulin-binding site, five residues.

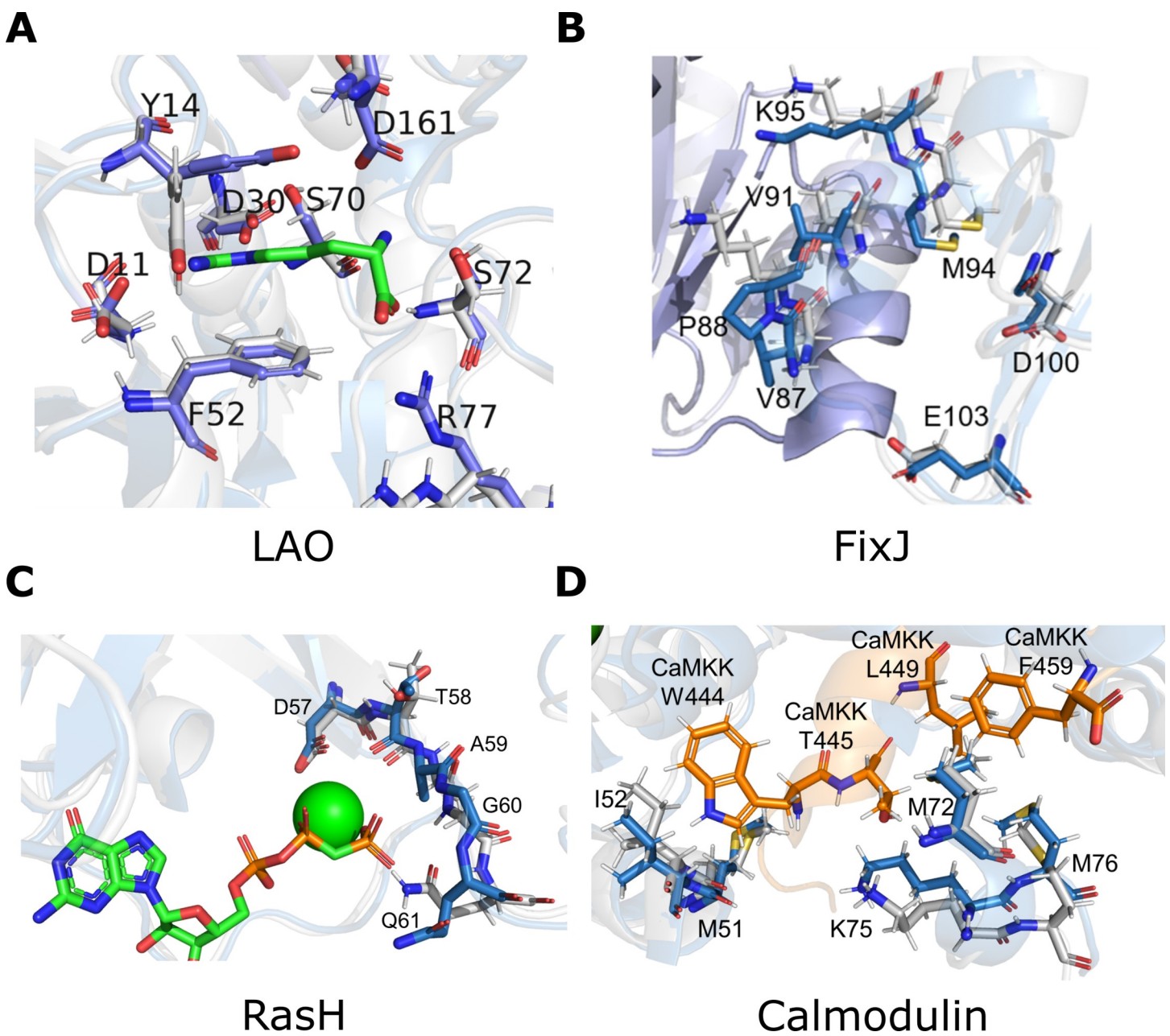

**Fig 8. 3D representation of binding sites.** (**A**) The native structure of LAO in the closed state (PDB ID: 1LAF) is depicted in blue and a protein designed with ResCue is shown in grey. The ligand arginine is shown as green sticks. (**B**) FixJ in the phosphorylated state (PDB ID: 1D5W) is depicted in blue and a protein designed with ResCue is shown in grey. Residues crucial for dimerization are shown in stick representation. (**C**) Native structure of RasH (PDB ID: 6Q21) with bound GTP (green sticks) is depicted in blue and a protein designed with ResCue is shown in grey. (**D**) Native structure of calmodulin (PDB ID: 1CKK) with bound peptide CaMMK (orange sticks). The Ca2$^+$ bound state is depicted in blue and a protein designed with ResCue is shown in grey. For all four protein designs, the ligand was not part of the starting structure.

of the conserved D54 residue induces the dimerization of the protein [31–34]. $res_{cc}^{20}(FixJ)$ consists of 20 residues that form an interaction network located at the dimerization interface (Figs 5B and 6B). The network includes the seven residues, which are critical for dimerization, namely V87, P88, V91, M94, K95, D100 and E103 [35]. Comparing the sequence logos (Fig 7B) of the five design approaches indicates that only ResCue sampled these seven amino acids correctly and highlights a major advantage of our approach: Since only one protein structure

was used during the design phase, all protocols were deprived of the interactions across the dimerization interface, which resulted in sequences probably unable to dimerize. Even SeqProf did not sample critical interface residues, because they are often less conserved [36] and depend on the occupancy of neighboring positions, which induces couplings. Thus, considering co-evolutionary constraints during the design process leads to favorable residue combinations, even without explicit knowledge of restraints related to dimerization (Fig 8B).

### RasH switches between two states for signal transduction

RasH is part of a signal transduction crucial for cell growth and differentiation. RasH adopts an 'off' and an 'on' state induced by a substantial conformational change in the so-called switch I region (residues 30–38) and switch II region (residues 60–76) [37]. The $res_{cc}^{20}(RasH)$ network connects both regions spanning residues 32–40 and 56–78. The networks highlight how the conformational shift is the product of the subtle interdependencies of protein residues (Figs 5C and 6C). The analysis of sequence logos determined for five GTP binding residues reveals that all four methods recover three residues well (T58, A59, G60) (Fig 7C). The other two (D57, Q61) were only recovered by ResCue. Superimposing the native structure with a model determined for a ResCue design confirms the correct orientation of the side chain residues (Fig 8C).

### The conformational switch of the calcium-binding messenger protein calmodulin

Calmodulin is an intermediate calcium-binding messenger protein, playing a critical role in coupling transient Ca2$^+$ influx to events in the cytosol and therefore the calcium signal transduction pathway [38, 39]. The protein undergoes a substantial conformational shift to bind the calmodulin-binding domain of the calmodulin-dependent protein kinase kinase (CaMKK) [40]. 25 residues (positions 34–53 and positions 72–76) are functionally relevant and the network is located at the N-terminal hydrophobic pocket anchoring T444 of the CaMKK peptide (Figs 5D and 6D). Due to the large number of functionally relevant residues, five exemplary residues were chosen with approximately similar distance; these were M51, I52, M72, K75, M76. The five residues are part of the coupling network and crucial for the CaMKK binding. The results highlight how our co-evolutionary approach biases the design towards the native amino acids (Fig 7D). Furthermore, sequences designed with co-evolutionary information sample realistic side-chain conformations, even in the absence of CaMKK in the design process (Fig 8D).

### Pros and cons of ResCue

In this work, we tested our hypothesis that incorporating co-evolutionary information into protein design helps Rosetta to design stable and functional proteins. For a benchmark consisting of ten difficult cases, native sequence recovery values and sequence similarity values were superior to three alternative design protocols. We imply that considering these evolutionary restraints helps to maintain function by recovering the couplings between residues.

ResCue outperforms SeqProf, because many crucial residues are not prominent in MSAs, and depend on the occupancy of neighboring positions. The design with RECON on only two conformations may underestimate its performance, since it scales well with the amount of different structures available for a given protein [11]. At the same time, the choice to use two conformations highlights how our approach is especially useful for proteins with little available additional structural information. Interestingly, ResCue compromises the energy score of the

designed sequences less than RECON MSD. Since the score is meant to reflect the thermodynamic stability of the design, impacting it to a high degree would possibly result in a less stable protein. The high native sequence recovery of ResCue, the fact that native sequences are close to optimal to their structure [13], and the experimental evidence that coupling guided design can generate novel sequences with stability similar to the wild type [41–43] suggests that the slightly increased Rosetta energy is thermodynamically acceptable. A restriction of our method is the need for an MSA consisting of *N* homologous sequences. However, given that more than 120 million protein sequences are deposited in the UniProt Databank [44] compared to the 163,141 structures deposited at the PDB [45], we suppose that ResCue can be applied to a wide variety of proteins. Our results are in agreement with earlier findings [20, 41] indicating that the integration of co-evolutionary information promotes stability and function in protein design.

Primary motivation of ResCue is to demonstrate, that co-evolutionary information can be leveraged for a Rosetta design algorithm that yields more natural sequences while conserving couplings that, at least in some cases, will be critical for plasticity or function, i.e. properties of the protein that are 'hidden' in a single structure/sequence pair. These couplings would encode required properties of the sequence but cannot be derived from a single conformation and in the absence of all binding partners. The Rosetta scoring function will optimize thermodynamic stability [6] of a single conformation but miss other aspects of plasticity or function. The altered scoring method with ResCue can be exploited 1) to explore a sequence space more similar to native sequences, 2) for conservative re-engineering while keeping known and unknown functions intact, or similar 3) to design on one structural conformation while not destabilizing other possible conformations. Alternative design protocols that can indirectly inform the design process about pairwise residue couplings were discussed: These are firstly, the design on multiple conformations (RECON), and secondly, design with a sequence profile. Amongst these, ResCue generates sequences most similar to the wild type and conserves the most plausible functionally important residues and couplings. Nonetheless, future experimental studies are required to confirm the suitability of our approach for the different suggested design scenarios.

It was previously shown that selecting a small number of mutations based on conservation information in sequence alignments can improve expression rates and predict improved protein stability [14]. This approach however failed to allow mutations in proximity of binding partners and co-factors to prevent activity loss in the first place. By leveraging co-evolutionary information, ResCue is going beyond the task of thermodynamic stabilization and can be exploited to re-design proteins including its functionally relevant sites, even when properties crucial for function are not encoded in one sequence/structure pair.

## Methods

### Collection of the benchmark *bench*<sub>*coev*</sub>

When compiling the benchmark, a major goal was to represent a wide variety of small to massive conformational shifts and proteins of different length *N*. Proteins were collected that exhibited conformational changes with the criteria that at least two conformations of the protein were known. To prevent discrimination of non-ResCue design protocols caused by low quality protein structures, only structures with an experimental resolution of at least 3Å were accepted. The existence of at least 10 x *N* non-redundant, homologous sequences was confirmed and the sequences were compiled to an MSA (see below). Sequences were considered redundant if they shared more than 80% sequence identity to the native sequence. All structural models were relaxed by means of Rosetta. For all single state design protocols, both structures were used as starting points and the results were pooled.

## GREMLIN-based co-evolution analysis

To analyze co-evolution between residues, multiple sequence alignments (MSA) were created using HHblits (E-value cutoff: 1.0e-10, Iterations: 4) [46, 47]. On average, the MSAs consisted of 24,700 sequences. We omitted sequences that did not cover at least 75% of the original sequence length. Additionally, we removed positions in the MSA with more than 75% gaps. The python version of GREMLIN was used to analyze each MSA and to create a tensor storing covariance values $DC_{i,j}(aa_x, aa_y)$ for all possible residue combinations $aa_x$, $aa_y$ at all positions $i$, $j$. The coupling strengths $DC_{i,j}(aa_x, aa_y)$ from the Markov Random Field (MRF) tensor were used to restrain designs with ResCue. MRF-values were preferred over the derived GREMLIN (pseudo)log-likelihood values [23] for two reasons: Firstly, (pseudo)log-likelihood values combine coupling strength and amino acid preferences at a certain position. In the MRF tensor, both values are listed separately. Here, we wanted to focus on coupling strengths. If desired, the user can utilize the already established FavorSequenceProfile term in addition to the ResCue coupling weight to incorporate a sequence conservation term (present in the log-likelihood). Secondly, usage of coupling strengths allows favoring correlations and penalizing anti-correlations. In contrast, the (pseudo)log-likelihood values reflect absolute coupling strengths and thus ignores this information. Eq 1 indicates how $DC_{i,j}(aa_x, aa_y)$ values were combined to deduce a coupling constraint $cc(i)$ for each single residue position $i$:

$$cc(i) = \sum_{i,j!=i} DC_{i,j}(aa_x, aa_y) \tag{1}$$

Here, and in all other formulae, $N$ is the length of the protein. The coupling strength $cs(seq)$ of a given sequence seq was determined by adding the $N$ $cc(i)$ values:

$$cs(seq) = \sum_{i=1}^{N} cc(i) \tag{2}$$

To assess the coupling recovery of a designed protein *prot*, the coupling recovery score *crs (prot)* was deduced from the *cs* values related to the designed and native sequences $seq_{Design}$ and $seq_{Native}$:

$$crs(prot) = \frac{cs(seq_{Design})}{cs(seq_{Native})} \tag{3}$$

## Assessment of native sequence recovery and sequence similarity

The native sequence recovery $nsr(seq_{Design})$ of a sequence $seq_{Design}$ is the fraction of residues $seq_{Design}[i]$ that match the corresponding native residues $seq_{Native}[i]$:

$$nsr(seq_{Design}) = \frac{1}{N} \sum_{i=1}^{N} ident(seq_{Design}[i], seq_{Native}[i]) \tag{4}$$

The binary function *ident*() determines the identity of two residues $aa_k$ and $aa_l$:

$$ident(aa_k, aa_l) = \begin{cases} 1 & if\ aa_k == aa_l \\ 0 & else \end{cases} \tag{5}$$

Analogously, sequence similarity *seqsim*(*seq*<sub>Design</sub>) was computed:

$$seqsim(seq_{Design}) = \frac{1}{N}\sum_{i=1}^{N} BLOSUM62(seq_{Design}[i], seq_{Native}[i]) \tag{6}$$

Scores for the similarity of corresponding residue pairs ware taken from the BLOSUM62 matrix [48]. All computations were performed using Biopython [49].

## Protein design with ROSETTA

The ROSETTA software suite was used for all different design approaches. RosettaScript XML files and commands can be found in S2 Supplement. Designs with no additional constraints (RoSSD) were performed by one round of fixed backbone rotamer optimization followed by repacking. RECON multistate designs utilized four rounds of fixed backbone design and a convergence step, as described in [8]. For each design, a PSSM was created by means of PSI-BLAST [24]. The PSSM was needed for the SeqProf RosettaScripts mover and the $MSA_{prot}$ served GREMLIN to deduce for each design the coupling tensor required for the ResCue Mover. At least 100 designs were generated for each protein in the benchmark and each approach. The resulting designs were scored with the ref 2015 Rosetta energy function.

## Network analysis of highly coupled residues

Regions of highly coupled residues were analyzed by using a similar technique, as described in [25]. First, for each residue $i$ of the native sequence the $cc(i)$ score (Eq 1) was determined. Then, a sliding window (window size of ten, step size of one) was used to identify regions containing highly co-evolving residues indirectly connected by slightly weaker coupled residues. To analyze regions with the highest co-evolutionary significance, we took the $res_{cc}^{20}(prot)$ residues and further analyzed how exactly they are coupled with each other. The networks were visualized with Cytoscape [50] and mapped on the protein structure by means of PyMOL [51]. Sequence logos were created with WebLogo [52].

## Supporting information

**S1 Supplement. Detailed description of benchmark proteins and its functionally relevant residues.**
(PDF)

**S2 Supplement. RosettaScript XML files and commands.**
(PDF)

**S3 Supplement. ResCue design full length web logos.**
(PDF)

**S4 Supplement. Coupling strength visualization of functionally relevant residues.**
(PDF)

## Author Contributions

**Conceptualization:** Samuel Schmitz, Moritz Ertelt, Jens Meiler.

**Data curation:** Samuel Schmitz, Moritz Ertelt.

**Formal analysis:** Samuel Schmitz, Moritz Ertelt.

**Funding acquisition:** Jens Meiler.

**Investigation:** Samuel Schmitz, Moritz Ertelt.

**Methodology:** Samuel Schmitz, Moritz Ertelt.

**Project administration:** Samuel Schmitz, Moritz Ertelt.

**Resources:** Jens Meiler.

**Software:** Samuel Schmitz, Moritz Ertelt.

**Supervision:** Samuel Schmitz, Rainer Merkl, Jens Meiler.

**Validation:** Samuel Schmitz, Moritz Ertelt.

**Visualization:** Samuel Schmitz, Moritz Ertelt.

**Writing – original draft:** Samuel Schmitz, Moritz Ertelt.

**Writing – review & editing:** Samuel Schmitz, Moritz Ertelt, Rainer Merkl, Jens Meiler.

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
