## [Decision Letter · Decision Letter 0]

11 Aug 2020

Dear Mr. Schmitz,

Thank you very much for submitting your manuscript "Rosetta Design with Co-Evolutionary Information Retains Protein Function" for consideration at PLOS Computational Biology.

As with all papers reviewed by the journal, your manuscript was reviewed by members of the editorial board and by several independent reviewers. In light of the reviews (below this email), we would like to invite the resubmission of a significantly-revised version that takes into account the reviewers' comments.

We cannot make any decision about publication until we have seen the revised manuscript and your response to the reviewers' comments. Your revised manuscript is also likely to be sent to reviewers for further evaluation.

Sincerely,

Anders Wallqvist

Associate Editor

PLOS Computational Biology

Arne Elofsson

Deputy Editor

PLOS Computational Biology

Reviewer's Responses to Questions

**Comments to the Authors:**

Reviewer #1: This manuscript describes the implementation and computational evaluation of an approach for incorporating pairwise residue correlations into fixed-backbone protein design. The concept is not new but the details of the method and the incorporation into the widely used Rosetta software suite should have value for protein engineers working on systems where flexibility and/or residue coupling are thought to be important. I have some concerns with the evaluation of the method, detailed below. I also feel that the manuscript would be much stronger if it included experimental validation of even a small number of designed sequences. A demonstration of a case where ResCue succeeded in preserving structure/stability/function while the other methods did not would go a long way to establishing the utility of the method. As it is, the authors have not clearly demonstrated that the approach is superior to a simple constraint that favors the native sequence.

Specifically, the sequence recovery and similarity metrics reward proximity to the native sequence, which can be trivially achieved by simply copying the starting sequence. This copy-native design algorithm would be superior to fixed-backbone design and the other 2 algorithms in all four of the metrics shown in Figure 3 (native sequence recovery, native sequence similarity, native cluster recovery, coupling recovery) and superior to ResCue in three of the four (with a pretty similar score of 100% for the fourth, coupling recovery). It's hard to say how copy-native would perform in the rosetta energy test-- presumably it will be worse unless there is a sampling problem identifying the native sequence. A more interesting and useful test would be to do a comparison with a simple variant of the default, fixed-backbone method ('RoSSD') with a single change: addition of a favor-native sequence bonus. The weight on the favor-native bonus could be tuned to give the same total sequence recovery as for ResCue, and then the two really interesting comparisons would be (1) the rosetta energy test (Fig. 2), and (2) the coupling recovery score. Presumably ResCue will give better coupling recovery (since it is one of the energy terms being directly optimized in ResCue designs), but I wonder how much better? And what will the Rosetta energy comparison look like?

I am also curious to understand how the cs(seq) term compares with the total log-likelihood of seq in the GREMLIN graphical model. It seems like they are similar, except maybe that cs(seq) only has the pairwise terms, not the one-body (single-position) terms that are also present in the log-likelihood. This makes me wonder how ResCue would compare to designing with the full GREMLIN (pseudo)log-likelihood as the additional energy term, rather than the pairwise interactions...

It wasn't completely clear which of the two structures was used in each benchmark case.

"Note, that the crs value can be larger than 100% which would suggest that the designed sequences fulfill additionally constraints not found in the native sequences, but in homologs" -- it's also worth pointing out here that the crs is directly being optimized as part of the energy function during sequence optimization, so the fact that it improves relative to the native is not so surprising. More interesting would be to evaluate coupling recovery in some way that isn't directly part of the energy function.

" in spacial proximity "

"and a seqsim value of 2.72" should be -2.72; also it would be helpful if "seqsim" were defined where first used...

"Rash H," should be "Ras H"

"Analyzing how different design approaches sample five exemplary residues" -- how were these 5 residues chosen?? Where subsets of positions have to be chosen for visual evaluation, perhaps full sequence logos could be provided in the supplementary material to reassure readers that the authors aren't cherry-picking.

Reviewer #2: This paper introduces a new design protocol into the Rosetta modeling package. The purpose of the method is to redesign existing protein structures while maintaining evolutionary couplings observed in a multiple sequence alignment. The method is compared to several earlier methods for using information in a multiple sequencing alignment to bias protein design.

The main strength of the paper are that the method is successful at biasing the designed sequences to preserve the observed evolutionary couplings. This is shown overall (Fig 4) and with a detailed analysis of specific cases. Furthermore, this is achieved with (what the authors consider to be) only a small change in the overall score metric used to design the proteins. The authors also show that evolutionary couplings are not recovered with methods that purely bias design based on residue frequencies in multiple seuqence alignments. Because I expect this method will be useful to the community of protein designers, I recommend publication following revisions.

In addition to this strength, there are also several weaknesses in the manuscript that should be addressed in resubmission.

First, the authors do not address the tradeoff between recovering couplings and optimizing their physical energy function in a series way. They write: "As noted above, it is essential, to balance carefully between having an efficient coupling restraint and designing physically realistic sequences as dictated by the Rosetta energy function." However, as far as I can tell they don't describe at all how they achieved this balance, or how we would know that the balance is good. Fig 2 shows that their method achieves better Rosetta total energies compared with the RECON and SeqProf methods, but their energies are still significantly higher than the RoSSD method (naive design without bias). How are we supposed to judge whether this difference in score is a significant cost? What efforts were made to balance their bias with the physical elements in the score function? One potential way to look at this would be to show what the total energy scores of the native proteins are before design, to show how much improvement is lost by enforcing the couplings.

Second, it is unclear how exactly this method would be used. Throughout the paper, native sequence recovery is referred to as a positive metric, and the authors write that increased nsr is a goal of their method: "Our ResCue design showed a significant increase both in the nsr and the seqsim

183 values, which were 70% and -1.20, respectively. Compared to the other design

184 approaches, these improvements were statistically significant (MW p < 5.0e-04 for nsr

185 and seqsim).". But of course it is trivial to get 100% native sequence recovery by avoiding design and outputing the starting sequence. So native sequence recovery can't be the only goal of the work. Could the authors be specific about what they are trying to optimize and how their method helps?

Two small comments to improve the paper:

-For benchmark proteins, could the paragraph "To study the coupling networks in detail, we present results for four benchmark proteins, LAO, FixJ, RasH, and calmodulin, which we have chosen for the following reasons:" go in the same odrer as the actual following information? Calmodulin is described first in the summary paragraph but last in the text.

-For Fig 7, please show the weblogo from the natural MSA as well as for the designed sequences. Also, these weblogos don't really help illustrate coupling. In addition to the weblogos, it would be good to show NxN matrices (where N is the number of positions) illustrating the level of mutual information (or some other metric?) between each of the positions in the weblogos. This would clearly show that coupling is present in the natural MSA, and whether it is present or absent in the four different methods.

Reviewer #3: In this article, Schmitz et al. develop a novel computational approach to perform protein design with Rosetta using evolutionary coupling information. The method is different from current approaches, and it is thoroughly and carefully benchmarked. The manuscript is well written and organized.

I have the following major questions / comments:

1) The exact intended use case for this method is a bit unclear. Is the method intended to improve protein thermostability, but for dynamic proteins? The manuscript can be strengthened by emphasizing the practical utility of the method more explicitly.

2) If improving thermostability is the goal, ResCue is making the rosetta energy worse, which indicates the protein is being destabilized. This major caveat wasn't sufficiently addressed, and as the authors themselves note in the final sentence of the main text: "experimental studies are required to finally confirm the suitability of our approach."

2) Also, if stabilizing the protein is the goal, it is unclear why preserving evolutionary couplings is so essential. Since the ResCue method requires a high resolution crystal structure of the target protein, functional residues are presumably known and do not need to be inferred from a MSA. The current state-of-the-art method for protein stabilization using MSAs and Rosetta is PROSS from Sarel Fleishman's group. While PROSS is not aware of evolutionary coupling, it has been demonstrated with experimental methods to stabilize proteins while preserving their function. Would it be fair to compare ResCue to PROSS?

**Have all data underlying the figures and results presented in the manuscript been provided?**

Reviewer #1: Yes

Reviewer #2: Yes

Reviewer #3: Yes

PLOS authors have the option to publish the peer review history of their article (what does this mean?). If published, this will include your full peer review and any attached files.

Reviewer #1: No

Reviewer #2: No

Reviewer #3: No
---

## [Decision Letter · Decision Letter 1]

28 Nov 2020

Dear Mr. Schmitz,

We are pleased to inform you that your manuscript 'Rosetta Design with Co-Evolutionary Information Retains Protein Function' has been provisionally accepted for publication in PLOS Computational Biology.

Best regards,

Anders Wallqvist

Associate Editor

PLOS Computational Biology

Arne Elofsson

Deputy Editor

PLOS Computational Biology

Reviewer's Responses to Questions

**Comments to the Authors:**

Reviewer #1: My concerns have been addressed

Reviewer #2: The authors have revised the manuscript and addressed the criticisms to my satisfaction.

Reviewer #3: revisions are acceptable

**Have all data underlying the figures and results presented in the manuscript been provided?**

Reviewer #1: Yes

Reviewer #2: Yes

Reviewer #3: Yes

PLOS authors have the option to publish the peer review history of their article (what does this mean?). If published, this will include your full peer review and any attached files.

Reviewer #1: No

Reviewer #2: No

Reviewer #3: No

---

## [Editor Report · Acceptance letter]

30 Dec 2020

PCOMPBIOL-D-20-01047R1 

Rosetta Design with Co-Evolutionary Information Retains Protein Function

Dear Dr Schmitz,

I am pleased to inform you that your manuscript has been formally accepted for publication in PLOS Computational Biology. Your manuscript is now with our production department and you will be notified of the publication date in due course.

With kind regards,

Livia Horvath
